# Radiofrequency Ablation of Painful Spinal Metastasis: A Systematic Review

**DOI:** 10.3390/curroncol32060301

**Published:** 2025-05-23

**Authors:** Jacopo Scaggiante, Salvatore Marsico, Andrea Alexandre, Simona Gaudino, Monica Ferrante, Riccardo Caronna, Ettore Squillaci, Iacopo Valente, Giuseppe Garignano, Francesco D’Argento, Reade De Leacy, Alessandro Pedicelli

**Affiliations:** 1Dipartimento di Diagnostica per Immagini, Radioterapia Oncologica ed Ematologia, Fondazione Policlinico “A. Gemelli” IRCCS, 00168 Rome, Italy; andrea.alexandre@policlinicogemelli.it (A.A.); simona.gaudino@policlinicogemelli.it (S.G.); monica.ferrante01@icatt.it (M.F.); riccardo.caronna01@icatt.it (R.C.); iacopo.valente@policlinicogemelli.it (I.V.); giuseppe.garignano@policlinicogemelli.it (G.G.); francesco.dargento@policlinicogemelli.it (F.D.); alessandro.pedicelli@policlinicogemelli.it (A.P.); 2Unità Operativa Complessa di Diagnostica per Immagini e Radiologia Interventistica, Ospedale Isola Tiberina—Gemelli Isola, 00186 Rome, Italy; ettore.squillaci@fbf-isola.it; 3Department of Radiology, Hospital del Mar, 08003 Barcelona, Spain; smarsico@hmar.cat; 4Department of Medicine and Life Sciences, Universitat Pompeu Fabra, 08002 Barcelona, Spain; 5Department of Neurosurgery, Mount Sinai Medical System, New York, NY 10029, USA; reade.deleacy@mountsinai.org

**Keywords:** vertebral bone metastasis, spinal neoplasms, spinal metastasis, radiofrequency ablation

## Abstract

Objective: To systematically evaluate the effectiveness and safety of radiofrequency ablation (RFA) for managing pain caused by spinal metastases. This review aimed to consolidate evidence on RFA’s analgesic efficacy and associated risks to inform clinical practice in palliative cancer care. Methods: A systematic review adhering to PRISMA guidelines was conducted. Databases were searched for studies evaluating RFA for spinal metastases pain. Inclusion criteria specified: randomized or non-randomized studies (prospective/retrospective); ≥3 adult patients; RFA used alone or combined with other treatments; reported pre- and post-RFA pain assessments; English language publication. Data extracted included patient demographics, primary tumor type, lesion location, pain scores (e.g., NRS/VAS), and complications. Pain response was assessed using definitions including the International Consensus Pain Response Endpoints (ICPRE) and definitions for moderate (≥2-point reduction) and high (≥4-point reduction) effectiveness. Results: This review included 33 studies, totaling 1336 patients (53.7% female) and 1787 treated lesions. The majority (85%) of studies reported highly effective pain management (≥4-point pain score reduction). The remaining 15% showed moderate effectiveness (≥2-point reduction). All studies reported achieving at least a partial pain response per ICPRE criteria. Mean pain scores decreased significantly from baseline (7.56/10) within 24–72 h (3.65) and remained low at 4 weeks (2.99), 12 weeks, and 24 weeks (both 2.70). Common primary cancers were lung (27.6%), breast (26.2%), and genitourinary (11.3%). Lesions were primarily in the thoracic (47.9%) and lumbar spine (47.3%). Crucially, no life-threatening (grade IV–V) complications occurred. The overall rate of grade I–III complications was low at 2.11%. Limitations: This systematic review is limited by its study-level nature, preventing detailed subgroup analyses regarding specific metastasis characteristics or the impact of complementary therapies. Conclusions: This systematic review suggests that RFA is a safe and effective treatment for pain control in patients with spinal metastases. It provides both rapid (within 24 h) and durable mid-term (up to 24 weeks) analgesia. The favorable safety profile, with a low complication rate, supports RFA as a valuable complimentary option within the multidisciplinary palliative management of painful spinal secondary tumors. Future randomized-controlled studies may help to further define its role when associated with other treatments.

## 1. Introduction

Spinal metastases represent a significant global health burden, affecting up to a third of cancer patients and causing severe pain and neurological complications [1]. While systemic therapies improve survival, pain management, neurological function preservation, and spinal stability remain crucial palliative goals [2]. Spinal metastases can lead to a variety of debilitating complications, including cord compression, pathological fractures, and radiculopathy, significantly impacting patients’ quality of life. Managing these complications often requires a complex, multidisciplinary approach involving oncologists, neurosurgeons, radiation oncologists, pain specialists, and rehabilitation professionals. Timely intervention is crucial to prevent irreversible neurological damage and improve patient outcomes [3].

Traditional management strategies, such as surgery and radiotherapy, while effective in certain cases, are associated with substantial morbidity, including infection, bleeding, and neurological deficits. These complications can significantly impact patients’ functional status and quality of life, often requiring prolonged hospital stays and rehabilitation.

Radiofrequency ablation (RFA) has emerged as a minimally invasive complementary technique to surgery and radiotherapy, as outlined by CIRSE guidelines [4], which offers a less invasive approach for the management of spinal metastases that can potentially reduce the need for more invasive procedures and their associated complications.

While individual studies have explored RFA, the existing literature still lacks a comprehensive, up-to-date synthesis that consolidates findings on its overall analgesic effectiveness and safety profile, particularly concerning short- and mid-term outcomes across diverse patient populations and tumor types. 

Therefore, the object of this study was to systematically evaluate the effectiveness and safety of radiofrequency ablation (RFA) for managing pain caused by spinal metastases.

This review aimed to consolidate evidence on RFA’s analgesic efficacy and associated risks to inform clinical practice in palliative cancer care, addressing the aforementioned gap by providing a robust overview of the current evidence.

## 2. Methods

### 2.1. Study Design and Research Strategy

We performed a systematic review and study-level meta-analysis of prospective and retrospective studies evaluating RFA techniques in the treatment of vertebral bone metastases according to the Preferred Reporting Items for Systematic Reviews and Meta-Analyses (PRISMA) guidelines [5].

Based on PRISMA guidelines, a comprehensive systematic search of PubMed, Embase, and Cochrane Library databases was conducted up to December 2023 with different combinations of the following search terms: “vertebral bone metastasis” or “spinal neoplasms” or “spinal metastasis” or “radiofrequency ablation” or “microwave ablation” or “cryoablation”.

We obtained additional articles using reference lists of articles identified in the initial searches.

### 2.2. Study Selection

Two independent reviewers (J.S.; M.F.) screened titles and abstracts of the retrieved articles for eligibility based on predefined inclusion criteria. Disagreements were resolved through discussion or by a third reviewer (A.P.). Full-text articles of potentially eligible studies were obtained and independently assessed by two reviewers against the inclusion criteria.

### 2.3. Inclusion Criteria

Studies were included if they met the following criteria:Randomized or non-randomized studies with at least 3 patients (prospective or retrospective);Adult patients with spinal metastases;Radiofrequency ablation (RFA) used alone or combined with other treatments;Reported pre- and post-RFA pain assessments;Published in English.

### 2.4. Data Extraction

A standardized data extraction form was developed to collect relevant information from included studies. Data extracted included study design, sample size, patient demographics (age, sex), tumor characteristics (primary site, histology), RFA technique, pain scores before and after RFA, follow-up duration, and adverse events. The pain scores extracted were the Numeric Rating Scale (NRS) [6] and Visual Analogue Score (VAS) [7] for back and leg pain. The Joanna Briggs Institute (JBI) Critical Appraisal Checklist for Case Series was used for critical appraisal (Appendix A).

### 2.5. Pain Response Definitions

According to the International Consensus Pain Response Endpoints (ICPRE) [8], pain response after radiofrequency ablation was defined as Complete Response, Partial Response, Pain Progression, and Indeterminate Response.

Moderate effective pain management was defined by a ≥2-point pain reduction from baseline to the last follow-up.

Highly effective pain management was defined by a ≥4-point pain reduction from baseline to the last follow-up.

### 2.6. Adverse Events Grading and Classification

Adverse events were extracted and classified according to the Common Terminology Criteria for Adverse Events (CTCAE) [9,10] in Mild events (Grade 1), Moderate events (Grade 2), Severe events (Grade 3), Life-threatening events (Grade 4), and Death related to adverse events (Grade 5).

### 2.7. Statistical Analysis

Pain response of the studies was analyzed as a continuous variable, with the effectiveness of the RFA treatment method evaluated and interpreted with the corresponding 95% confidence interval. Data from the two pain scales (VAS and NRS) were pooled for analysis as both scales were reported as an 11-point system ranging from 0 to 10. Classic χ^2^ test, *Q*^2^*,* and *I*^2^ statistics were used to assess the existence and magnitude of between-study heterogeneity. The significance level was set at 0.05. All analyses were conducted by both fixed-effect and random-effect models to increase robustness. The random-effects model was employed for the analysis. The studies in the analysis were assumed to be a random sample from a universe of potential studies, and this analysis was used to make an inference to that universe [11].

All data analyses were conducted and tested by both R (Version 3.4.1) and Comprehensive Meta-Analysis Version 4.

## 3. Results

### 3.1. Study Characteristics

The initial systematic search yielded 234 studies, from which 46 total duplicates were removed.

Then, 123 articles were removed after title and abstract screening, and another 32 studies were excluded because they did not meet the inclusion criteria after reading the full text.

The final database included 33 case-series studies from 2004 to 2023 involving a total of 1336 patients with 1787 lesions treated with RFA (Figure 1).

Twenty-eight articles had a retrospective design and five were prospective studies.

Sample sizes ranged from 3 to 206 patients.

Of the included studies, twelve were authored by US-based researchers, one by Canadian researchers, eleven by European researchers, and nine by Chinese researchers.

### 3.2. Patient Characteristics (Table 1)

Among 1336 patients, 53.7% were female, and the mean age was 62.3 years old.

The most common primary tumor sites were the lung (27.6%), breast (26.2%), and genitourinary system (11.3%).

Other primary tumors sites were multiple myeloma (3.0%), sarcomas (3.9%), thyroid (2.5%), prostate (4.5%), pancreaticobiliary system (1.2%), liver (2.4%), melanoma/skin (2.1%), lymph node (0.6%), head/neck (0.4%), colon/GI system (7.8%), oral squamous cell (0.1%), peripheral nerve sheath tumor (0.2%), myogenic hemangioendothelioma (0.1%), others (5.9%), and unknown/noncancerous origin (0.2%).

The majority of lesions were located in the thoracic spine (47.9%), followed by the lumbar spine (47.3%) and sacral spine (4.5%).

**Table 1 curroncol-32-00301-t001:** Study characteristics.

Authors	Study Design	Country	Year	N Tumors	N Patients
Madaelil TP et al. [12]	Retrospective	USA	2016	16	11
Greenwood TJ et al. [13]	Retrospective	USA	2015	36	21
Prezzano KM et al. [14]	Retrospective	USA	2019	28	26
Wallace AN et al. [15]	Retrospective	USA	2015	72	
Tomasian A et al. [16]	Retrospective	USA	2021	266	166
Levy J et al. [17]	Prospective	USA	2020	134	100
Ragheb A et al. [18]	Retrospective	USA	2022	50	23
Bagla S et al. [19]	Prospective	USA	2016	69	50
Sayed D et al. [20]	Prospective	USA	2019	34	30
Levy J et al. [21]	Prospective	USA	2023	206	206
Reyes M et al. [22]	Retrospective	USA	2018	72	49
Anchala PR et al. [23]	Retrospective	USA	2014	128	92
Lane MD et al. [24]	Retrospective	CANADA	2011	53	36
Senol N et al. [25]	Retrospective	EUROPE	2022	56	41
Pusceddu C et al. [26]	Retrospective	EUROPE	2021	41	35
Sandri A et al. [27]	Retrospective	EUROPE	2010	11	11
Maugeri R et al. [28]	Retrospective	EUROPE	2017	18	18
Giammalva GR et al. [29]	Retrospective	EUROPE	2022	63	54
Masala S et al. [30]	Retrospective	EUROPE	2004	3	3
Alfonso M et al. [31]	Retrospective	EUROPE	2023	26	14
Hoffmann RT et al. [32]	Retrospective	EUROPE	2008	18	15
Shawky Abdelgawaad A et al. [33]	Retrospective	EUROPE	2021	75	60
Pusceddu C et al. [34]	Retrospective	EUROPE	2023	21	16
Madani K et al. [35]	Retrospective	EUROPE	2022	24	18
Lv N et al. [36]	Retrospective	CHINA	2020	47	35
Zhou X et al. [37]	Retrospective	CHINA	2021	6	6
Zheng L et al. [38]	Retrospective	CHINA	2014	38	26
Wang F et al. [39]	Retrospective	CHINA	2021	17	15
Han X et al. [40]	Retrospective	CHINA	2021	23	23
Zhang C et al. [41]	Prospective	CHINA	2020	15	15
Tian QH et al. [42]	Retrospective	CHINA	2022	51	51
He Y et al. [43]	Retrospective	CHINA	2021	19	19
Lu CW et al. [44]	Retrospective	CHINA	2019	51	51

### 3.3. Outcomes: Pain Response (Table 2)

All the 33 included studies (100%) achieved Partial Response according to ICPRE criteria.

Also, 28 out of 33 articles (85%) reported highly effective pain management (≥4-point reduction on a pain scale) after RFA, while the remaining 5 studies (15%) reported moderate pain relief (≥2-point pain reduction).

Using the 11-point system, the mean pain value was 7.56 at baseline, 3.65 at 24–72 h post-intervention, 2.99 at 2–4 weeks, 2.69 at 12 weeks, and 2.70 at 24 weeks.

**Table 2 curroncol-32-00301-t002:** Pain response.

	Pre-Intervention				24–72 h Post-Intervention				2–4 Weeks Post-Intervention				12 Weeks Post-Intervention				24 Weeks Post-Intervention			
Authors	Mean	95% C.I.	S.E.	Z-Value	Mean	95% C.I.	S.E.	Z-Value	Mean	95% C.I.	S.E.	Z-Value	Mean	95% C.I.	S.E.	Z-Value	Mean	95% C.I.	S.E.	Z-Value
Madaelil TP et al. [12]	8.00	**-**	**-**	**-**	**-**	**-**	**-**	**-**	3.00	**-**	**-**	**-**	**-**	**-**	**-**	**-**	**-**	**-**	**-**	**-**
Greenwood TJ et al. [13]	8.00	7.25–8.75	0.38	20.87	-	-	-	-	2.9	1.822–3.978	0.55	5.273	-	-	-	-	-	-	-	-
Prezzano KM et al. [14]	4.52	3.38–5.66	0.58	7.791	-	-	-	-	2.7	1.626–3.774	0.548	4.927	1.6	0.637–2.563	0.491	3.256	-	-	-	-
Wallace AN et al. [15]	8.00	7.56–8.44	0.22	35.728	-	-	-	-	2.9	2.207–3.593	0.354	8.202	-	-	-	-	-	-	-	-
Tomasian A et al. [16]	8.00	7.88–8.12	0.06	130.476	-	-	-	-	3.00	2.88–3.12	0.061	48.929	3.00	2.88–3.12	0.061	48.929	-	-	-	-
Levy J et al. [17]	8.20	7.91–8.49	0.15	55.836	-	-	-	-	3.9	3.392–4.408	0.259	15.049	3.7	3.209–4.191	0.251	14.769	3.5	2.958–4.042	0.276	12.661
Ragheb A et al. [18]	6.90	-	-	-	3.50	-	-	-	2.80	-	-	-	-	-	-	-	0.70	-	-	-
Bagla S et al. [19]	5.90	-	-	-	-	-	-	-	-	-	-	-	2.1	-	-	-	-	-	-	-
Sayed D et al. [20]	5.77	-	-	-	-	-	-	-	3.33	-	-	-	2.64	-	-	-	2.61	-	-	-
Levy J et al. [21]	7.80	7.57–8.03	0.12	65.854	-	-	-	-	4.7	4.318–5.082	0.195	24.092	3.6	3.19–4.01	0.209	17.223	3.2	2.79–3.61	0.209	15.31
Reyes M et al. [22]	7.90	7.32–8.48	0.30	26.813	3.5	2.899–4.101	0.306	11.422	-	-	-	-	-	-	-	-	-	-	-	-
Anchala PR et al. [23]	7.51	7.08–7.94	0.22	34.539	-	-	-	-	2.25	1.827–2.673	0.216	10.433	-	-	-	-	1.75	1.296–2.204	0.232	7.557
Lane MD et al. [24]	7.20	6.64–7.76	0.29	25.08	-	-	-	-	1.68	-	-	-	-	-	-	-	-	-	-	-
Senol N et al. [25]	7.40	-	-	-	2.50	-	-	-	2.50	-	-	-	-	-	-	-	3.20	-	-	-
Pusceddu C et al. [26]	5.70	5.21–6.19	0.25	22.811	-	-	-	-	0.9	0.686–1.114	0.109	8.233	-	-	-	-	3.00	2.878–3.122	0.062	48.023
Sandri A et al. [27]	8.00	-	-	-	1.80	-	-	-	1.90	-	-	-	-	-	-	-	-	-	-	-
Maugeri R et al. [28]	8.50	-	-	-	3.50	-	-	-	2.80	-	-	-	2.60	-	-	-	3.00	-	-	-
Giammalva GR et al. [29]	7.81	-	-	-	5.20	-	-	-	4.10	-	-	-	3.50	-	-	-	-	-	-	-
Masala S et al. [30]	8.60	-	-	-	2.60	-	-	-	-	-	-	-	-	-	-	-	-	-	-	-
Alfonso M et al. [31]	7.70	6.97–8.43	0.37	20.664	-	-	-	-	2.6	1.985–3.215	0.314	8.286	-	-	-	-	-	-	-	-
Hoffmann RT et al. [32]	8.50	-	-	-	5.50	-	-	-	-	-	-	-	3.50	-	-	-	-	-	-	-
Shawky Abdelgawaad A et al. [33]	7.20	6.68–7.72	0.27	27.11	-	-	-	-	-	-	-	-	3.00	2.525–3.475	0.242	12.372	-	-	-	-
Pusceddu C et al. [34]	8.00	-	-	-	-	-	-	-	0.5	-	-	-	0	-	-	-	-	-	-	-
Madani K et al. [35]	7.30	6.34–8.26	0.49	14.901	-	-	-	-	2.00	0.00	-	-	-	-	-	-	-	-	-	-
Lv N et al. [36]	7.52	7.11–7.93	0.21	35.802	2.79	2.638–2.942	0.773	36.089	2.14	2.026–2.254	0.058	36.678	-	-	-	-	2.23	2.098–2.362	0.067	33.235
Zhou X et al. [37]	7.60	5.82–9.38	0.91	8.386	3.2	1.8–4.6	0.714	4.479	3.5	2.156–4.844	0.686	5.103	-	-	-	-	-	-	-	-
Zheng L et al. [38]	7.69	7.33–8.05	0.18	42.325	6.62	6.296–6.944	0.165	40.008	3.62	3.308–3.932	0.159	22.771	2.77	2.509–3.031	0.133	20.824	2.96	2.667–3.253	0.149	19.833
Wang F et al. [39]	8.46	8.06–8.86	0.21	41.037	1.73	1.269–2.191	0.235	7.354	2.24	1.936–2.544	0.155	14.431	1.83	1.488–2.172	0.175	10.48	1.86	1.489–2.231	0.189	9.832
Han X et al. [40]	7.39	6.83–7.95	0.29	25.869	-	-	-	-	4.52	3.882–5.158	0.325	13.896	-	-	-	-	2.3	1.622–2.978	0.346	6.645
Zhang C et al. [41]	7.86	7.43–8.30	0.22	35.397	-	-	-	-	-	-	-	-	3.51	2.842–4.178	0.341	10.299	-	-	-	-
Tian QH et al. [42]	7.43	7.00–7.86	0.22	34.013	2.25	1.879–2.621	0.189	11.902	1.96	1.568–2.352	0.2	9.788	1.91	1.529–2.291	0.195	9.813	1.86	1.484–2.236	0.192	9.696
He Y et al. [43]	7.19	6.259–8.121	0.475	15.14	4.39	3.787–4.993	0.307	14.28	2.89	2.274–3.506	0.314	9.195	1.75	1.273–2.227	0.243	7.196	-	-	-	-
Lu CW et al. [44]	8.07	-	0.79	-	4.61	-	0.75	-	4.38	-	0.61	-	-	-	-	-	4.34	-	0.31	-
Random	7.56	7.323–7.791	0.119	63.31	3.646	2.524–4.769	0.573	6.365	2.996	2.455–3.538	0.276	10.838	2.688	2.284–3.091	0.206	13.054	2.706	2.012–3.399	0.354	7.647
Prediction Interval	7.56	6.522–8.592			3.646	−0.49–7.782			2.996	0.583–5.41			2.688	1.217–4.159			2.706	0.038–5.373		

### 3.4. Complications

No life-threatening complications (grade IV–V) were reported in any of the included studies.

Grade 1 complications (i.e., asymptomatic or mild symptoms requiring observation only) occurred in 13/1280 patients (1.02%). Grade 2 (minimal, local, or noninvasive intervention indicated) occurred in 8/1280 patients (0.63%), while grade 3 complications (medically significant but not immediately life-threatening; hospitalization or prolongation of hospitalization indicated) occurred in 6/1280 patients (0.47%).

Overall complications included temporary post-procedural radicular pain, asymptomatic spinal cord edema, transient neurological motor deficits, pulmonary embolism, difficulty urinating, erectile dysfunction due to spinal cord venous infarct, transient pain following epidural leaks during treatment, drug hypersensitivity, folliculitis, hematoma, fluid collection, pneumonia, and respiratory failure.

## 4. Discussion

Painful spinal metastases present a significant challenge in oncologic care, burdening patients’ quality of life and functional status. The quest for effective and minimally invasive analgesic treatments is therefore paramount. This systematic review was conducted to synthesize the evidence on radiofrequency ablation (RFA).

Our findings suggest that RFA is a highly effective and safe modality for pain palliation in these patients. Specifically, this review of 33 studies encompassing 1336 patients demonstrated that 85% of studies reported highly effective pain management (a reduction of ≥4 points on a pain scale), and all included studies achieved at least a partial pain response according to ICPRE criteria. Furthermore, RFA was associated with a low overall complication rate of 2.11% (grade I–III), with no life-threatening (grade IV–V) complications reported. 

The analysis indicates RFA’s potential to achieve pain relief in the short to mid-term (from 48–72 h to 24 weeks), contributing to improved functionality and overall quality of life.

Mean pain scores decreased significantly from baseline (7.56/10) within 24–72 h (3.65) and remained low at 4 weeks (2.99), 12 weeks, and 24 weeks (2.70).

It is crucial to consider the results in the context of the limitations associated with conventional treatment modalities for spinal metastases. The ability to achieve pain management within 48–72 h represents a substantial clinical advancement, particularly in light of the 3–6 week timeframe typically associated with radiation therapy for pain palliation [45], suggesting that RFA can be effectively utilized either as a complementary treatment for pain control or as an alternative when radiotherapy has failed or is contraindicated.

This finding resonates with the OPuS One trial [21], the largest, prospective multicenter study that specifically evaluated RFA for metastatic lytic bone disease independent of radiation therapy and similarly found significant and durable pain reduction following the procedure.

The safety profile of RFA, as highlighted in our review, further strengthens its case for wider adoption. Indeed, we found a relatively low rate of complications. No life-threatening complications (grade IV–V) were reported, and the overall complication rate was 2.11% (grade I–III). These findings are consistent with previous studies reporting low complication rates (3.0%) associated with RFA for spinal metastases [16,46,47].

However, it is important to acknowledge that RFA is not without risk. Complications such as temporary post-procedural radicular pain, asymptomatic spinal cord edema, and transient neurological motor deficits were observed.

While RFA is generally considered safe, careful patient selection and meticulous technique are crucial to minimize the risk of complications. The safety profile of RFA can be further improved by combining it with vertebroplasty, which can reduce the risk of vertebral fracture [29]. Additionally, the use of bipolar RFA with increased target temperature has been shown to have an excellent safety profile and high rates of pain relief and local tumor control [6].

While our review predominantly included retrospective case series, the consistency of positive outcomes across these studies from different countries, coupled with the robust findings from prospective trials, provides evidence for the efficacy and safety of RFA. To further validate these findings and establish the long-term durability of pain relief, future research should prioritize prospective studies with larger sample sizes and extended follow-up periods. This includes exploring its role in combination with other therapies, such as radiotherapy, or against other minimally invasive techniques such as cryoablation and microwave ablation, to develop comprehensive and personalized treatment strategies.

### Limitations

This systematic review has limitations that reflect the heterogeneity across the included studies related to the inherent study-level nature of this systematic review. Indeed, we included a mix of retrospective and prospective studies and a wide range of sample sizes. Although this may have introduced biases into the analysis, we wanted to provide a comprehensive, up-to-date synthesis of the individual studies on this minimally invasive technique, which is widely used in different countries (USA, Canada, Europe, China), with data published since 2004 by Masala et al. [30]. 

Our review provides an overall assessment of RFA’s effectiveness and safety, but the variability in the source data restricted our ability to perform granular subgroup analyses that could offer deeper insights into specific patient populations or technical aspects. 

Specifically, the influence of metastasis characteristics, such as their lytic, blastic, or mixed nature, on the degree of pain relief and interventional success could not be definitively determined due to inconsistent reporting of these variables across the included studies. 

Likely, the variability of treated lesions reflects the interventionalists’ learning curve over decades. Nowadays, as reported by Tomasian et al. in 2018 [48], it is widely known that RFA is primarily used for the treatment of lesions that are mainly osteolytic due to their lower intrinsic impedance compared with sclerotic bone lesions, which allows the RF circuit to generate higher temperatures to ensure cell death.

Accordingly, details of different RFA techniques employed, such as the use of monopolar versus bipolar systems, specific ablation zone sizes, and peak ablation temperatures, were often not reported with sufficient detail to provide a patient-level meta-analysis, which was beyond the scope of the aggregated data available on the safety and effectiveness of RFA. Future research, ideally through large-scale prospective studies or registries with standardized data collection protocols, would be invaluable to address these specific questions and allow for more detailed subgroup analyses to further refine the optimal use of RFA for painful spinal metastases.

## Figures and Tables

**Figure 1 curroncol-32-00301-f001:**
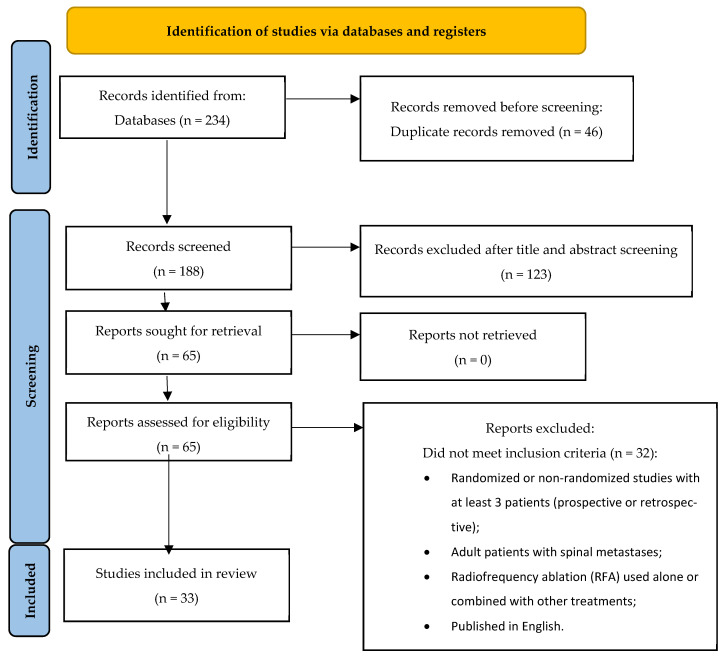
Study flow diagram.

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
