# Peer review of "Radiofrequency Ablation of Painful Spinal Metastasis: A Systematic Review"

_curroncol, 2025, doi:10.3390/curroncol32060301_

Round 1

Reviewer 1 Report

Comments and Suggestions for Authors

Thank you for asking me to review this manuscript. I have found this study interesting and well presented. However, I have two main comments for the authors.

  1. I would like to see an evaluation of the quality of the selected studies. The Joanna Briggs Institute (JBI) Critical Appraisal Checklist for Case Series could be applied.
  2. It is important to perform subgroup analyses between patients who receive RFA in combination with other procedures (what kind of procedures, etc.) and patients treated with RFA as a standalone procedure. Is there any difference?

Author Response

Reviewer comment 1: I would like to see an evaluation of the quality of the selected studies. The Joanna Briggs Institute (JBI) Critical Appraisal Checklist for Case Series could be applied.

Authors comment 1: Thank you for your suggestion. We agree with you that The Joanna Briggs Institute (JBI) Critical Appraisal Checklist for Case Series is indeed a solid tool to assess the methodological quality of a study. We reviewed the studies and modified the Data Extraction paragraph adding “The Joanna Briggs Institute (JBI) Critical Appraisal Checklist for Case Series was used for critical appraisal.”

Reviewer comment 2: It is important to perform subgroup analyses between patients who receive RFA in combination with other procedures (what kind of procedures, etc.) and patients treated with RFA as a standalone procedure. Is there any difference?

Authors comment 2: We agree with you that subgroup analyses would offer deeper insights into specific patient populations or technical aspects. Our review provides an overall assessment of RFA's effectiveness and safety, but the variability in the source data restricted our ability to perform granular subgroup analyses. Based on your point, we added a limitation paragraph to the article which read: “This systematic review has limitations that reflect the heterogeneity across the included studies related with the inherent study-level nature of the systematic review. Indeed, we included a mix of retrospective and prospective studies and a wide range of sample sizes. Despite this may introduce biases in the analysis, we wanted to provide a comprehensive, up-to-date synthesis of the individual studies on this minimally-invasive technique which is widely used in different countries (USA, Canada, Europe, China) with data published since 2004 by Masala et al. [30]. Our review provides an overall assessment of RFA's effectiveness and safety, but the variability in the source data restricted our ability to perform granular subgroup analyses that could offer deeper insights into specific patient populations or technical aspects.   Specifically, the influence of metastasis characteristics, such as their lytic, blastic, or mixed nature on the degree of pain relief and interventional success could not be definitively determined due to inconsistent reporting of these variables across the included studies.  Likely, the variability of treated lesions reflects the interventionalists learning curve over decades. Nowadays, as reported by Tomasian et al. in 2018 [48], it is overall known that RFA is primarily used for treatment of lesions that are mainly osteolytic because their lower intrinsic impedance compared with sclerotic bone lesions which allow the RF circuit to generate higher temperatures to ensure cell death.Accordingly, details of different RFA techniques employed, such as the use of monopolar versus bipolar systems, specific ablation zone sizes, peak ablation temperatures were often not reported with sufficient detail to provide a patient-level meta-analysis which was beyond the scope of the aggregated data available on safety and effectiveness of RFA. Future research, ideally through large-scale prospective studies or registries with standardized data collection protocols, would be invaluable to address these specific questions and allow for more detailed subgroup analyses to further refine the optimal use of RFA for painful spinal metastases.”

Reviewer 2 Report

Comments and Suggestions for Authors

You reviewed the relevant literature on the application of RFA in thoracolumbar metastases, correctly following the PRISMA protocol. You conclude on their efficacy in reducing back pain, both when applied alone or associated with other treatments.

Your systematic review is very similar to that of Murali et al. : by applying more stringent exclusion criteria, these Authors reached significantly different conclusions, highlighting low evidence of efficacy. 

I agree with Murali et al, for two essential reasons: - there are no randomized controlled clinical trials; - even in the references reported, radiofrequency represent more often an implementation of cementoplasty techniques and are used alone only rarely. Therefore, their real clinical efficacy cannot be completely and surely defined.
All these aspects should be included and highlighted in the text, modifying the conclusive aspects of this review

Author Response

Reviewer comment 1: You reviewed the relevant literature on the application of RFA in thoracolumbar metastases, correctly following the PRISMA protocol. You conclude on their efficacy in reducing back pain, both when applied alone or associated with other treatments.

Your systematic review is very similar to that of Murali et al. : by applying more stringent exclusion criteria, these Authors reached significantly different conclusions, highlighting low evidence of efficacy. 

I agree with Murali et al, for two essential reasons: - there are no randomized controlled clinical trials; - even in the references reported, radiofrequency represent more often an implementation of cementoplasty techniques and are used alone only rarely. Therefore, their real clinical efficacy cannot be completely and surely defined.
All these aspects should be included and highlighted in the text, modifying the conclusive aspects of this review

Authors comment 1: Thank you for your suggestions.  We agree that lack of randomized controlled trials represents a significant gap in literature. Based on your point, we added a limitation paragraph to the article which read: “This systematic review has limitations that reflect the heterogeneity across the included studies related with the inherent study-level nature of the systematic review. Indeed, we included a mix of retrospective and prospective studies and a wide range of sample sizes. Despite this may introduce biases in the analysis, we wanted to provide a comprehensive, up-to-date synthesis of the individual studies on this minimally-invasive technique which is widely used in different countries (USA, Canada, Europe, China) with data published since 2004 by Masala et al. [30]. 

Our review provides an overall assessment of RFA's effectiveness and safety, but the variability in the source data restricted our ability to perform granular subgroup analyses that could offer deeper insights into specific patient populations or technical aspects.   

Specifically, the influence of metastasis characteristics, such as their lytic, blastic, or mixed nature on the degree of pain relief and interventional success could not be definitively determined due to inconsistent reporting of these variables across the included studies.  

Likely, the variability of treated lesions reflects the interventionalists learning curve over decades. Nowadays, as reported by Tomasian et al. in 2018 [48], it is overall known that RFA is primarily used for treatment of lesions that are mainly osteolytic because their lower intrinsic impedance compared with sclerotic bone lesions which allow the RF circuit to generate higher temperatures to ensure cell death.

Accordingly, details of different RFA techniques employed, such as the use of monopolar versus bipolar systems, specific ablation zone sizes, peak ablation temperatures were often not reported with sufficient detail to provide a patient-level meta-analysis which was beyond the scope of the aggregated data available on safety and effectiveness of RFA. Future research, ideally through large-scale prospective studies or registries with standardized data collection protocols, would be invaluable to address these specific questions and allow for more detailed subgroup analyses to further refine the optimal use of RFA for painful spinal metastases.”

Round 2

Reviewer 1 Report

Comments and Suggestions for Authors

Thank you for asking me to review this manuscript. It is well written and interesting. 

According to my previous comment, in a systematic review the quality of the included studies is important. The purpose of this evaluation is to assess the methodological quality of a study and to determine the extent to which a study has addressed the possibility of bias in its design, conduct and analysis. The authors have added in the Data Extraction paragraph “The Joanna Briggs Institute (JBI) Critical Appraisal Checklist for Case Series was used for critical appraisal.” However, they omit to present the results of this appraisal. Please provide the results in details.

Author Response

Reviewer Comments1: 

"Thank you for asking me to review this manuscript. It is well written and interesting. 

According to my previous comment, in a systematic review the quality of the included studies is important. The purpose of this evaluation is to assess the methodological quality of a study and to determine the extent to which a study has addressed the possibility of bias in its design, conduct and analysis. The authors have added in the Data Extraction paragraph “The Joanna Briggs Institute (JBI) Critical Appraisal Checklist for Case Series was used for critical appraisal.” However, they omit to present the results of this appraisal. Please provide the results in details."

Authors response 1: Thank you for your suggestion. Please, see the Zip attachment including the JBI Critical Checklist for the included studies for your kind consideration.

Reviewer 2 Report

Comments and Suggestions for Authors

For changes made to the text and the limitations of the available data from the literature, it might be more appropriate to modify your conclusions and the abstract, by defining RFA as a safe technique with a very low risk of complications, especially in lytic metastases. But its real effectiveness cannot be correctly defined because the available data mainly refer to associated treatments. 

Author Response

Reviewer comment 1: "For changes made to the text and the limitations of the available data from the literature, it might be more appropriate to modify your conclusions and the abstract, by defining RFA as a safe technique with a very low risk of complications, especially in lytic metastases. But its real effectiveness cannot be correctly defined because the available data mainly refer to associated treatments. "

Authors response 1: Thank you for your suggestion. Based on your recommendation, we added a limitation paragraph to the abstract that read "This systematic review is limited by its study-level nature, preventing detailed subgroup analyses regarding specific metastasis characteristics or the impact of complementary therapies.". Additionally, we added a sentence to the abstract conclusion that read "Future randomized-controlled study may help to further define its role when associated with other treatments." As you mentioned, at the end of the conclusion paragraph in the main text we recommend "To further validate these findings and establish the long-term durability of pain relief, future research should prioritize prospective studies with larger sample sizes and extended follow-up periods. This includes exploring its role in combination with other therapies (...)".

Round 3

Reviewer 1 Report

Comments and Suggestions for Authors

Please include the method and results of the Trial appraisal checklist in the manuscript or send them as supplementary material.

I provide a paper as an example.
